# The Anti-Inflammatory and Antioxidant Effects of Sodium Propionate

**DOI:** 10.3390/ijms21083026

**Published:** 2020-04-24

**Authors:** Alessia Filippone, Marika Lanza, Michela Campolo, Giovanna Casili, Irene Paterniti, Salvatore Cuzzocrea, Emanuela Esposito

**Affiliations:** 1Department of Chemical, Biological, Pharmaceutical and Environmental Sciences, University of Messina, Viale Ferdinando Stagno D’Alcontres, 31-98166 Messina, Italy; afilippone@unime.it (A.F.); mlanza@unime.it (M.L.); campolom@unime.it (M.C.); gcasili@unime.it (G.C.); ipaterniti@unime.it (I.P.); salvator@unime.it (S.C.); 2Department of Pharmacological and Physiological Science, Saint Louis University, St. Louis, MO 63103, USA

**Keywords:** SCFAs, SP, NF-κB, inflammation, oxidative stress

## Abstract

The major end-products of dietary fiber fermentation by gut microbiota are the short-chain fatty acids (SCFAs) acetate, propionate, and butyrate, which have been shown to modulate host metabolism via effects on metabolic pathways at different tissue sites. Several studies showed the inhibitory effects of sodium propionate (SP) on nuclear factor kappa-light-chain-enhancer of activated B cells (NF-κB) pathway. We carried out an in vitro model of inflammation on the J774-A1 cell line, by stimulation with lipopolysaccharide (LPS) and H_2_O_2_, followed by the pre-treatment with SP at 0.1, 1 mM and 10 mM. To evaluate the effect on acute inflammation and superoxide anion-induced pain, we performed a model of carrageenan (CAR)-induced rat paw inflammation and intraplantar injection of KO_2_ where rats received SP orally (10, 30, and 100 mg/kg). SP decreased in concentration-dependent-manner the expression of cicloxigenase-2 (COX-2) and inducible nitric oxide synthase (iNOS) following LPS stimulation. SP was able to enhance anti-oxidant enzyme production such as manganese superoxide dismutase (MnSOD) and heme oxygenase-1 (HO-1) following H_2_O_2_ stimulation. In in vivo models, SP (30 and 100 mg/kg) reduced paw inflammation and tissue damage after CAR and KO_2_ injection. Our results demonstrated the anti-inflammatory and anti-oxidant properties of SP; therefore, we propose that SP may be an effective strategy for the treatment of inflammatory diseases.

## 1. Introduction

The Short-chain fatty acids (SCFAs) are carboxylic acids defined by the presence of an aliphatic tail of two to six carbons. Although SCFAs can be produced naturally through host metabolic pathways, the major site of production is the colon requiring the presence of specific colonic bacteria [1]. SCFAs levels may be increased by the frequent introduction of fiber-rich products into the diet. Acetate (C2), propionate (C3), and butyrate (C4) are the major SCFAs released through fermentation of fiber and resistant starches [2]. In particular, in the intestine, SCFAs exert a trophic effect on the intestinal epithelium and play a key role in the modulation of colonic blood flow, gastrointestinal (GI) motility, and fluid and electrolyte absorption [3]. Several studies focused their attention on the effect of SCFAs on inflammatory signaling pathways, and it was well demonstrated that butyrate inhibits nuclear factor kappa-light-chain-enhancer of activated B cells (NF-κB) translocation, cytokines production and prevents oxidative damage in a murine model of nephropathy and colitis [4,5,6]. Therefore, there is one report that SCFAs are able to inhibit the DNA binding and NF-κB-mediated transcription of inflammatory markers after IFN-γ-stimulation of RAW 264.7. ERK1/2 signaling pathway are involved in the potential anti-inflammatory effects of the SCFAs. Moreover, recent evidence suggested the potential therapeutic applications of butyrate in the treatment of metabolic and inflammatory diseases [7]. Based on this finding, the present study aimed to investigate the mechanism of action of sodium propionate (SP), focusing our attention not only on inflammation but also on the modulation of oxidative stress and pain. The peculiarity of our study is to provide original data about the mechanisms that undergoing SP with in vitro and in vivo experimental models that has not previously been done before. The study was dividing into two steps: the first was to identify the molecular mechanism of SP by two different in vitro models by stimulating murine macrophages cell line with liposaccharide (LPS) to induce inflammation and with hydrogen peroxide (H_2_O_2_) to induce oxidative stress. The second step was to study SP mechanism in vivo, including pain modulation, in a model of acute inflammation induced by CAR and in a model of superoxide anion-induced pain induced by of potassium peroxide (KO_2_).

## 2. Results

### 2.1. In Vitro Studies

#### 2.1.1. Effect of SP on Vitality Following LPS Stimulation

In order to choose the highest SP concentrations with the lowest toxicity, cell viability was assessed stimulating J774-A1 with different concentrations (0.1, 10, 100 µM, and 1, 10 mM) of SP. Treatment of SP at different concentration such as 100 μM, 1 mM, and 10 mM, markedly increased the basal proliferation of cells. Then, we decided to use the concentrations of 0.1–1 and 10 μM. (Figure 1A). Moreover, we assessed the viability following LPS stimulation. The J774-A1 cells pre-treated with SP showed an increased proliferation following LPS-induced cytotoxicity (Figure 1B).

#### 2.1.2. Effect of SP on the Expression of iNOS and COX-2 Following LPS Stimulation

To evaluate the nitrosative stress and lipid peroxidation induced by LPS 10 μg/mL stimulation and the protective role of SP, we evaluated inducible nitric oxide synthase (iNOS) and cicloxigenase-2 (COX-2) expressions by western blot analysis. Basal levels of iNOS were observed in the control groups, whereas LPS stimulation induced a significant increase in iNOS expression (aaa *p* < 0.001 versus Ctr, Figure 2A,A1). Pre-treatment with SP reduced the expression of iNOS in a concentration-dependent manner, significant at 1 μM and 10 μM. COX-2 was significantly increased after LPS stimulation, whereas pre-treatment with SP, for all the concentrations, significantly reduced COX-2 expression (Figure 2B,B1).

#### 2.1.3. Effect of SP on the Expression of IκBα and NF-κB following LPS Stimulation

To investigate the molecular mechanism of SP against LPS-induced inflammation, we evaluated NF-κB pathway. Basal levels of IκBα was detected in control groups, while LPS stimulation induced IκBα degradation. Treatment with SP, for all three concentrations, restored IκBα expression (Figure 2C,C1). LPS stimulation induced NF-κB translocation into the nucleus, while SP treatment at all concentrations significantly reduce NF-κB translocation (Figure 2D,D1).

#### 2.1.4. Anti-Oxidant Effect of SP in J774-A1 Cell Cultures Stimulated with H_2_O_2_

To evaluate the antioxidant effect of SP and its potential capability to induce recovery after oxidative stress, J774-A1 cells were pre-treated with SP and then stimulated with H_2_O_2_ 200 µM for 10 min. We observed that cytotoxicity induced by H_2_O_2_ decreased the cell viability about 80%, while the pre-treatment with SP at the concentrations of 1 μM and 10 μM significantly restored cell viability, highlighting its potential anti-oxidant effect (Figure 3).

#### 2.1.5. SP Reduces the Nitrite Production and MDA Level in J774-A1

We also tested lipid peroxidation through the production of malondialdehyde (MDA) in LPS-stimulated macrophages to verify the anti-inflammatory activity of SP; moreover, nitrite production was measured because NO is a toxic molecule released by the innate immune cells during disease. The control groups released low levels of NO_2_^−^; instead, H_2_O_2_ stimulation significantly increased NO_2_^−^ production. Pre-treatment with SP in a concentration dependent-manner significantly decreased the NO_2_^−^ levels (μM/mL) (Figure 4A). Phospholipids membrane are susceptible to the attack of free radicals during oxidative stress; for this reason, we evaluated by MDA assay, the lipid peroxidation of the membrane. Our data showed a significant increase of MDA level after H_2_O_2_ stimulation while pre-treatment with SP attenuated the levels at a concentration of 1 μM and 10 μM (Figure 4B).

#### 2.1.6. Effect of SP on Anti-Oxidant Enzymes In Vitro

The stimulation with H_2_O_2_ caused important oxidative damage that reflected in a modified expression of anti-oxidant enzymes such as MnSOD and HO-1. By Western blot analysis, our results showed a significant increase in both enzymes following H_2_O_2_; however, SP at a concentration of 10 μM up-regulated their expression (Figure 5A,A1,B,B1).

### 2.2. In Vivo Studies

#### 2.2.1. Effect of SP on Time-Course of CAR-Induced Paw Edema in Rats

Injection of CAR into the hind paw rapidly induced a clear and time-dependent increase in paw edema volume. That increase started at 3 h until 6 h (Figure 6). A significant reduction of paw edema volume was observed in rats treated with SP at 10 mg/kg, 30 mg/kg and 100 mg/kg compared to the CAR group (*p*-value of less than 0.05 was considered significant) (Figure 6).

#### 2.2.2. Histological Analyses of Paw Tissues and MPO Activity in CAR-Treated Rats

Histological evaluation was performed by hematoxylin and eosin (H&E) staining as described above. No histological damage was observed in control rats (Figure 7A), whereas important damage was observed 6 h after CAR injection with a marked accumulation of infiltrating inflammatory cells, edema and loss of normal muscle paw architecture (Figure 7B; see histological score Figure 7F), compared to control (Figure 7A; see histological score Figure 7F). SP treatment significantly reduced morphological alterations (Figure 7C–E). Moreover, histological damage was associated with an increased neutrophil infiltration as shown by MPO activity. SP 30 mg/kg and 100 mg/kg significantly reduced the enzyme activity (Figure 7G).

#### 2.2.3. Effect of SP on the Expression of iNOS and COX-2 in Hind Paw Tissue from CAR-Treated Rats

We also determined the effect of SP (10 mg/kg, 30 mg/kg and 100 mg/kg) on pro-inflammatory enzymes as COX-2 and iNOS. COX-2 and iNOS expressions were assessed by western blot analysis paw homogenates 6 h after CAR injection. The expression of COX-2 was increased in paw tissues subjected to CAR injection compared to control rats. On the other hand, COX-2 expression was decreased by oral treatment with SP at 30 mg/kg and 100 mg/kg (Figure 8A,A1). Moreover, Figure 8B showed a significant increase in iNOS expression in the CAR group, which was significantly reduced following the treatment with SP at 30 mg/kg and 100 mg/kg (Figure 8B,B1).

#### 2.2.4. Effect of SP on Time-Course of O_2_^−^ Anion-Induced Inflammatory Pain

Intraplantar injection of O_2_^−^ evoked an increase in paw edema of rapid onset (within 5 min) that reached a peak at 60 min. Oral administration of SP at 10, 30 and 100 mg/kg reduced paw edema compared to the vehicle within 1 h (Figure 9A).

#### 2.2.5. The Nociceptive Response Following O_2_^−^ Intraplantar Injection

The analgesic response was measured at 30 min and 60 min after O_2_^−^ intraplantar injection. SP administration at doses of 10, 30, and 100 mg/kg at the time point of 30 min did not give evidence of analgesic activity. Moreover, at 60 min after O_2_^−^ intraplantar injection, the oral administration of SP at all doses significantly increased the latency before tail curling in rats (Figure 9B).

#### 2.2.6. Analgesic Effect of SP Evaluated by Formalin Test

The early analgesic response occurs 5 min after the formalin injection (a *p* < 0.005 versus. Sham) (phase I). On this phase, oral treatment of SP (10–30 mg/kg and 100 mg/kg) did not evoke an analgesic response associated with damage induced by O_2_^−^ intraplantar injection (Figure 9C); in phase II, formalin response following O_2_^−^ injection, was significantly blocked by SP administration at the dose of 30 mg/kg and 100 mg/kg (Figure 9D).

#### 2.2.7. Histological Evaluation of Paw Tissues and MPO Activity following O_2_^−^ Intraplantar Injection

No histological damage was observed in control rats (Figure 10A), whereas damage was observed 1 h after O_2_^−^ injection associated to inflammatory cells infiltration, edema, and alteration of tissue architecture (Figure 10B) compared to control group (Figure 10A). SP treatment at the all doses (10–30 mg/kg and 100 mg/kg) (Figure 10C–E; see histological score Figure 10F) significantly reduced morphological alterations. However, the administration of SP (10–30 mg/kg and 100 mg/kg) significantly reduced MPO activity in a dose-dependent manner (Figure 10G).

#### 2.2.8. SP Reduce O_2_^−^ Induced Oxidative Stress

It is well known that O_2_^−^ intraplantar injection modulates the expression of anti-oxidant enzyme as MnSOD and HO-1 in rats hind paw [8]. As shown in Figure 11, SP treatment (30 mg/kg and 100 mg/kg) significantly up regulated the expression of these enzymes compared to the control group (Figure 11A,B).

#### 2.2.9. SP Reduce GSH Levels in Rat Paws Following O_2_^−^ Intraplantar Injection

GSH is produced in tissues as a result of GSH peroxidase activity and it is immediately reduced by GSH reductase, thereby maintaining constant levels of reduced glutathione in the normal tissue. Basal levels of GSH were measured in control rats, while O_2_^−^ intraplantar injection contributes to increased GSH in levels. Oral administration of SP at concentrations of 30 mg/kg and 100 mg/kg, significantly increased GSH levels compared to the control group. However, 10 mg/kg SP did not show any significant modulation (Figure 11C).

## 3. Discussion

Inflammation and oxidative stress are linked together in a large number of pathophysiological processes. [9,10]. SCFAs exerted different anti-inflammatory properties, mainly derived from the inhibition of NF-κB translocation. To date, butyrate was exhaustively studied among the SCFAs, while the molecular mechanism of propionate is not well documented [11]. A recent study demonstrated that propionate, similarly to butyrate, may inhibit the activation of NF-κB in colon adenocarcinoma cell line [5] with a consequent reduction of pro-inflammatory factors expression such as TNF-α and IL-1β in colon tissues. Moreover, we have previously shown the neuroprotective effect of SP in an in vitro neuroinflammatory model and in vivo model of spinal cord injury (SCI), recognizing in SP an optimal therapeutic target for neuroinflammatory disorders [12]. In this study, we confirmed the ability of SP to contain the peripheral acute inflammation trough the down-regulation of NF-κB pathway. Moreover, we demonstrated that SP was able to reduce oxidative stress and nociceptive stimuli. It is well known that LPS stimulation induces the nuclear translocation of NF-κb [13] and the subsequent degradation of Iκb-α, as well as the stimulation of iNOS and COX-2 expression [14]. Here, the pre-treatment with SP at the concentration of 1 µM and 10 µM on macrophages cells significantly decreased the activation of inflammatory modulators induced by NF-κB activation. In recent years, many studies evidenced that oxidative stress plays a crucial role in the development and propagation of inflammation [15,16]. The production of various reactive oxidant species, in excess compared to an endogenous antioxidant defense mechanism, promotes the development of a state of oxidative stress with significant biological consequences [17]. H_2_O_2_ is a relatively stable oxidant but is also converted by neutrophils and macrophages to more reactive species such as superoxide and hydroxyl radicals. As these oxidants are detrimental to the surrounding cells, in vitro and in vivo studies tested many antioxidants compounds to improve anti-oxidant response. The in vitro study to test if SP treatment exerted antioxidant activity showed a key role in the up-regulation of the anti-oxidant enzymes Mn-SOD and HO-1 and in the reduction of MDA formation and NO_2_^−^ production. These results highlight that SP prevents oxidative stress by scavenging free radical species and/or boosting the endogenous antioxidant system capacity by stimulating the synthesis of endogenous antioxidants. The first conclusion resulted in in vitro studies, confirmed SP effect against inflammation, and showed for the first time its potential effect to contrast oxidative stress. 

To better evaluate the properties of SP in a more complex system, we developed an in vivo model of acute inflammation induced by CAR in rats. CAR intraplantar injection is one of the techniques used to screen anti-inflammatory drugs because it exerted profound development of edema and cytokines release [18]. We evidenced that SP possessed the ability to counteract peripheral inflammation following CAR-injection, in particular, decreasing paw edema, reducing histological damage, and infiltrating inflammatory cells. Moreover, in an in vivo model, SP was able to prevent the activation of NF-κB pathway. The peculiarity of this study was to go in deep into the anti-oxidant and analgesic effect of SP. Through the in vivo model, we injected a superoxide anion donor, KO_2_, used as a generator of superoxide anion and considered a critical mediator in nociception [19]. KO_2_ has been used as a generator of superoxide anion in vivo [8]. Superoxide anion is a precursor of additional reactive oxygen species (ROS) with enhanced toxicity, including hydroxyl radical, hypochlorous acid, and singlet oxygen [20,21]. The role of superoxide anion is not directly related to its deleterious effects but rather to its activation of signaling molecules. For instance, the capability of neuropathy and inflammation-induced superoxide anion is to activate kinases and transcription factors such as NF-κB which guide the increase of the production of pro-inflammatory molecules [22,23]. 

Surprisingly, we elucidated by behavioral nociceptive tests the analgesic effects of oral administration of SP showing the ability to attenuate the pain following superoxide intraplantar injection. Moreover, as described by Wang [24], superoxide intraplantar injection modified hind paw tissue architecture and increased oxidative stress in situ. We showed that SP oral treatment was able to preserve paw tissue lost. the anti-oxidant properties of SP was evidenced by the significant up regulated of anti-oxidant enzymes MnSOD and HO-1 that was accompanied by the reduction of GSH levels, highlighting the key role of SP in maintaining antioxidant defense systems.

## 4. Materials and Methods

The murine macrophage cell line J774-A1 and the culture Dulbecco’s Modified Eagle’s Medium (DMEM) were obtained from ATCC^®^ (Manassas, VA, USA). Fetal bovine serum (FBS), Bio-Rad protein assay, 3-(4,5-dimethylthiazol-2-yl)-2,5-diphenyl tetrazolium bromide (MTT), SP, LPS, H_2_O_2_ and anti-laminin A/C antibody were obtained from Sigma-Aldrich (Saint Louis, Missouri, USA). Anti-NF-κB, anti-IκBα, anti-COX-2, anti-MnSOD, anti-HO-1, and anti β-actin antibody for Western blot analysis were obtained from Santa Cruz Biotechnology (Dallas, TX, USA). Peroxidase-conjugated anti-mouse secondary antibody, peroxidase-conjugated goat and anti-rabbit IgG were obtained from Jackson Immuno Research (West Grove, PA, USA). All compounds used in in vivo study were purchased from Sigma-Aldrich Company Ltd. (Poole, United Kingdom). All solutions used for in vivo infusions were prepared using non-pyrogenic saline (0.9% *wt*/*vol* NaCl; Baxter Healthcare Ltd., Thetford, United Kingdom).

### 4.1. In Vitro Studies

#### 4.1.1. Murine Macrophage Cell Cultures and Treatments

The murine macrophage cell-line J774-A1 (ATCC^®^ TIB-67) was cultured in 75 cm^2^ flasks in a complete medium composed by DMEM with the addition of 10% of (FBS). Cells were maintained at 37 °C and 5% CO_2_. For the cell viability of J774-A1, 4 × 10^4^ cells were plated (in a volume of 150 µL) in 96-well plates. Thereafter, the medium was replaced with fresh medium and cells were treated with six different concentrations (0.1, 1, 10, 100 μM and 1, 10 mM) of SP, to determine the concentrations with less toxicity. In the first set of the experiment to test inflammatory properties of SP, we pre-treated cells with SP at the concentrations of 0.1, 1, 10, 100 μM and after 2 h we stimulated cells with LPS (10 µg/mL) [25]. After 24 h, we performed mitochondria-dependent dye for live cells (tetrazolium dye; MTT) to formazan [26] and Western blot analysis for NFκB, IκB-a, iNOS, and COX-2. In the second set of the experiment to test the anti-oxidant properties of SP, cells were pre-treated with SP at the concentrations of 0.1, 1, 10, 100 μM and then were stimulated with H_2_O_2_ 200 µM for 10 min. After this experimental time, we performed the MTT assay and the Western blot analysis for HO-1 and MnSOD.

#### 4.1.2. Western Blot Analysis

Western blot analysis was performed as previously described [27]. J774-A1 cells were washed twice with ice-cold phosphate-buffered saline (PBS), harvested, and resuspended in Tris-HCl 20 mM pH 7.5, NaF 10 mM, 150 µl NaCl, 1% Nonidet P-40 and protease inhibitor cocktail (Roche, Switzerland). After 40 min cell lysates were centrifuged at 12,000 rpm for 15 min at 4 °C. Protein concentration was estimated by the Bio-Rad protein assay using bovine serum albumin as standard. Samples were then heated at 95 °C for 5 min and equal amounts of protein separated on a 10–15% SDS-PAGE gel and transferred to a PVDF membrane (Immobilon-P). The following primary antibodies were used: anti-NF-κB (1:500; Santa Cruz Biotechnology, Dallas, TX, USA sc 8008), anti-IκBα (1:500; Santa Cruz Biotechnology, Dallas, TX, USA), anti-iNOS (1:1000 BD transduction), anti-COX-2 (1:500; Santa Cruz Biotechnology, sc 376861), anti-MnSOD (1:500; Santa Cruz Biotechnology, Dallas, TX, USA) and anti-HO-1 (1:500; Santa Cruz Biotechnology, Dallas, TX, USA). Antibody dilutions were made in PBS/5% *w*/*v* nonfat dried milk/0.1% Tween-20 (PMT) and membranes incubated overnight at 4 °C. Membranes were then incubated with secondary antibody (1:2000, Jackson ImmunoResearch, West Grove, PA, USA) for 1 h at room temperature. To ascertain that blots were loaded with equal amounts of proteins, they were also incubated in the presence of the antibody against β-actin protein (cytosolic fraction 1:500; Santa Cruz Biotechnology, Dallas, TX, USA) or lamin A/C (nuclear fraction 1:500 Sigma–Aldrich Corp.). Signals were detected with enhanced chemiluminescence (ECL) detection system reagent according to the manufacturer’s instructions (Thermo Fisher Scientific, MA, USA). The relative expression of the protein bands was quantified by densitometry with BIORAD ChemiDoc TMXRS+ software and standardized to β-actin and lamin A/C levels. Data are representative of at least three replicates.

#### 4.1.3. NO_X_ Assay

NO_X_ levels were measured in the J774-A1 cell line supernatant as described by Talero et al. [28]. Briefly, the supernatant was incubated with nitrate reductase (670 mU/mL) and β-nicotinamide adenine dinucleotide 3′-phosphate (NADPH) (160 μM), at room temperature for 3 h. The total nitrite concentration in the supernatant was then measured using the Griess reaction, by adding 100 µL of Griess reagent (0.1% *w*/*v*) naphthyl-ethylen-diamide-dihydrochloride in H_2_O and 1% (*w*/*v*) sulphanilamide in 5% (*v*/*v*) concentrated H_3_PO_4_; vol. 1:1 to the 100 µL sample. The optical density at 550 nm (OD550) was measured using a microplate reader.

#### 4.1.4. Malondialdehyde (MDA) Assay

J774-A1 cells (1 × 105 cells/well) were seeded in poly-l-lysine-coated six-well plates. The cells were harvested to detect the levels of malondialdehyde (MDA) using the thiobarbituric acid (TBA) reactive substances assay (TBARS assay kit, Cayman Chemical, Ann Arbor, Michigan MI, USA, Item NO. 10009055). Briefly, TBA was added to samples and standard and incubated at 95 °C for 60 min, cooled in ice bath for 10 min. Then, samples were transferred to microplate and analyzed with microplate reader at OD of 532 nm.

#### 4.1.5. Data Analysis

All values are expressed as mean ± standard error of the mean (SEM) of “n” observations. Each analysis was performed three times with three samples replicates for each one. The results were analyzed by one-way analysis of variance (ANOVA) followed by a Bonferroni posthoc test for multiple comparisons. A *p*-value of less than 0.05 was considered significant.

### 4.2. In Vivo Studies

#### 4.2.1. Animals

The study was carried out by using Sprague–Dawley male rats (200–230 g, Envigo, RMS Srl Udine, Italy). Food and water were available ad libitum. This study was approved by the University of Messina Review Board for the care of animals, in compliance with Italian regulations on protection of animals (n° 399/2019-PR released on 05/24/2019). Animal care was in accordance with Italian regulations on the use of animals for the experiment (D.M.116192) as well as with EEC regulations (O.J. of E.C. L 358/1 12/18/1986).

#### 4.2.2. Carrageenan-Induced Paw Edema

Paw edema was induced by a subplantar injection of CAR (100 µL of a 1% suspension in 0.85% saline). Changes in paw volume were measured as previously described [29] using a plethysmometer (Ugo Basile, Varese, Italy) immediately before CAR injection, and, then, at hourly intervals for 6 h. Edema was expressed as the increase in paw volume (mL) after CAR injection relative to the pre-injection value for each rat. Results are reported as a paw-volume change (mL).

#### 4.2.3. Induction of edema by O_2_^−^

Lightly anesthetized rats (CO_2_ (80%)/O_2_ (20%)) received a sub-plantar injection of O_2_^−^ (1mM) or its vehicle in a total volume of 50 mL into the right hind paw as described by Salvemini et al. [30].

#### 4.2.4. Experimental Groups

Rats were divided in several groups:

Sham+ vehicle group: rats were orally administered with saline (*n* = 8);

Sham+ SP (10 mg/kg) group: rats were orally administered with SP at the dose of 10 mg/kg (*n* = 8);

Sham+ SP (30 mg/kg) group: rats were orally administered with SP at the dose of 30 mg/kg (*n* = 8);

Sham+ SP (100 mg/kg) group: rats were orally administered with SP at the dose of 100 mg/kg (*n* = 8);

Carrageenan group: rats were subplantar injected with CAR (*n* = 10);

Carrageenan+ SP (10 mg/kg): rats were orally administered with 10 mg/kg SP 30 min before CAR subplantar injection (*n* = 10);

Carrageenan+ SP (30 mg/kg): rats were orally administered with 30 mg/kg SP 30 min before CAR subplantar injection (*n* = 10);

Carrageenan+ SP (100 mg/kg): rats were orally administered with 100 mg/kg SP 30 min before CAR subplantar injection (*n* = 10);

O_2_^−^ group: rats were subplantar injected with O_2_^−^ (*n* = 10);

O_2_^−^ + SP (10 mg/kg): rats were orally administered with 10 mg/kg SP 30 min before O_2_^−^ injection (*n* = 10);

O_2_^−^ + SP (30 mg/kg): rats were orally administered with 30 mg/kg SP 30 min before O_2_^−^ injection (*n* = 10);

O_2_^−^ + SP (100 mg/kg): rats were orally administered with 100 mg/kg SP 30 min before O_2_^−^ injection (*n* = 10);

Furthermore, data regarding groups Sham+ SP 10 mg/kg, Sham+ 30 mg/kg, and Sham+ 100 mg/kg were not showed because SP administration did not demonstrate histological changes compared to Sham group. The doses of SP were based on a previous dose-response study made in our laboratory.

#### 4.2.5. Paw Edema Measurement

Changes in paw volume were measured as previously described [31]. Briefly, paw volume was measured with a plethysmometer (Ugo Basile, Comerio, Varese, Italy) immediately before the injection of O_2_^−^ (before 30 min) and after at 1 h. Edema was expressed as the increase in paw volume (milliliters) after O_2_^−^ injection relative to the pre-injection value for each animal. Results are expressed as paw volume change (milliliters).

#### 4.2.6. Behavioral Tests

##### Tail-Flick Test

Tail-flick test was performed in the following experimental groups: Sham+ vehicle group; O_2_^−^ group; O_2_^−^ + SP 10 mg/kg; O_2_^−^ + SP 30 mg/kg and O_2_^−^ + SP 100 mg/kg. Nociceptive testing was performed by placing the distal portion of the tail of each animal in a water bath maintained at 52 °C. The time latency to withdrawal the tail (tail-flick) was measured at different time points: 30 min before O_2_^−^ subplantar injection (pre-dose), 30 min and 60 min after O_2_^−^ subplantar injection. The determination of antinociception was assessed between 07:00 and 10:00.

##### Formalin Test

Formalin test was performed in the following experimental groups: Sham+ vehicle group; O_2_^−^ group; O_2_^−^ + SP 10 mg/kg; O_2_^−^ + SP 30 mg/kg and O_2_^−^ + SP 100 mg/kg. To evaluate prolonged noxious stimulus produced by formalin injection, the rats were lightly anesthetized and 50 µL of 5% formalin solution was injected subcutaneously (s.c.) into the dorsal surface of the right hind paw with a 30-g needle. The rat was then placed in an open plexiglass chamber with a mirror positioned on the opposite side to allow unhindered observation of the formalin-injected paw. Pain-related behavior was quantified by counting the incidence of spontaneous flinching/shaking of the injected paw. In the first phase (phase I), the total number of flinches/shakes × min was counted starting the formalin injection until 5 min after that; in the second phase (phase II) the total number of flinches/shakes was counted every 5 min until 60 min after formalin injection. After 1 h of observation, rats were sacrificed by anesthetic overdose.

#### 4.2.7. Myeloperoxidase Activity (MPO Activity)

MPO activity, an index of polymorphonuclear cell accumulation, was determined as previously described in the hind paw tissues of all experimental groups [31]. The rate of change in absorbance was measured spectrophotometrically at 650 nm. MPO activity was measured as the quantity of enzyme degrading 1 mM of peroxide min-1 at 37 °C and was expressed in units per gram weight of wet tissue.

#### 4.2.8. Histological Examination of the CAR-Inflamed Hind Paw

Biopsies of hind paws were taken 6 h following CAR injection. Histology was performed as previously described [31]. The degree of paw damage was evaluated according on a six-point score: 0 = no inflammation, 1 = mild inflammation, 2 = mild/moderate inflammation, 3 = moderate inflammation, 4 = moderate/severe inflammation and 5 = severe inflammation.

#### 4.2.9. Histological Examination of the O_2_^−^-Inflamed Hind Paw

For histopathological examination, hind paws were taken 60 min after the intra-plantar injection of O_2_^−^. Tissue from the pads of rat hind paws was removed with a scalpel and processed. [30].

#### 4.2.10. Western Blot Analysis for COX-2, iNOS, MnSOD and HO-1

Cytosolic and nuclear extracts of the hind paws were performed as previously described. The levels of COX-2, iNOS, MnSOD, and HO-1 were quantified in the cytosolic fraction. The filters were blocked with 1X PBS, 5% (*w*/*v*) nonfat dried milk for 40 min at room temperature and subsequently probed with one of the following primary antibodies (all from Santa Cruz Biotechnology, Dallas, TX, USA) COX-2 (1:500; Santa Cruz Biotechnology, Dallas, TX, USA), iNOS (1:500; Santa Cruz Biotechnology, Dallas, TX, USA), Mn-SOD (1:500; Santa Cruz Biotechnology, Dallas, TX, USA ), or HO-1 (1:500; Santa Cruz Biotechnology, Dallas, TX, USA) in 1X PBS, 5% *w*/*v* non-fat dried milk, 0.1% Tween-20 at 4 °C, overnight. Membranes were incubated with secondary antibody (peroxidase conjugated bovine anti-mouse IgG secondary antibody or peroxidase-conjugated antirabbit IgG, 1:2000; Jackson Immuno Research, West Grove, PA1:2000) for 1 h at room temperature. Bands were detected by chemiluminescence (ECL) system (Thermo, USA), visualized with the ChemiDoc XRS (Bio-Rad, Hercules, CA, USA) and analyzed by using Image Lab 3.0 software (Bio-Rad, Hercules, CA, USA). The expression levels of β-actin served as an internal control for protein loading. Data are representative of at least three replicates.

#### 4.2.11. Glutathione Assay (GSH Assay)

One hour after the O_2_^−^-intraplantar injection, rats were killed, and hind paws were collected and then suspended in 5 mL of 10 mM sodium phosphate buffer (pH 7.4) containing 8.56% (*w*/*v*) sucrose and homogenized in variable amounts of the buffer by using of an Ultra-Turax (Wilmington, NC, USA) tissue homogenizer. Reduced (GSH) and oxidized glutathione form (GSGG) ratio was measured using an enzymatic method that utilizes Ellman’s Reagent (DTNB) and glutathione reductase (GR). DTNB reacts with reduced glutathione to form a yellow product. The optical density was measured at 412 nm.

#### 4.2.12. Statistical Evaluation

All values in the figures and text are expressed as mean standard deviation (SD) of N observations. For in vivo studies, N represents the number of animals studied. In the experiments involving histology, the figures shown are representative of at least three experiments performed on different days. The results were analyzed by one-way ANOVA followed by a Bonferroni post hoc test for multiple comparisons.

## 5. Conclusions

Previously, it was shown that SP exerts beneficial effects on the intestinal epithelium, inhibiting inflammation and modulating oxidative stress in a dextran sulfate sodium (DSS)-induced colitis mouse model [2]. It is common to believe that SCFAs play a key role in reducing inflammation in disorders of the gastrointestinal tract because the SCFAs are already produced in the colonic lumen by anaerobic fermentation of carbohydrates, so the increased intake of these attenuates colon damages. Here, we elucidated that SP exerts important anti-inflammatory effects, accompanied by the ability to prevent oxidative stress and reduced pain also in peripheral tissue out from the origin area. The anti-inflammatory and anti-oxidant activities of SP are associated with its ability to down-regulate the NF-κB pathway and up-regulate antioxidant enzymes. Although there are limitations and challenges with animal models of acute pain modulation, clinically relevant models are essential for a fuller understanding of the behavioral consequences of acute and chronic pain and how they relate to the numerous complex histopathological cascades and to other inflammatory signaling pathways. This study is also unable to provide information about the inflammatory assessment of chronic pain. Our findings suggested the potential therapeutic efficacy of SP in several pathological events involving pro-inflammatory molecules and oxidants mediators recruitment; therefore, SP could represent a promising therapeutic target for the treatment of acute and chronic inflammatory diseases.

## Figures and Tables

**Figure 1 ijms-21-03026-f001:**
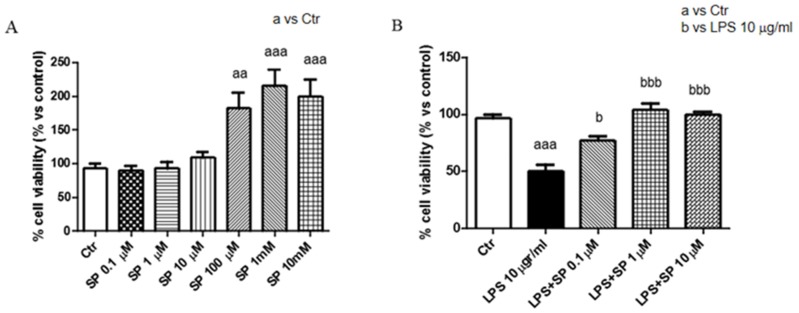
Anti-inflammatory effects of sodium propionate (SP) in cell-lines J774-A1 following liposaccharide (LPS) stimulation. Cell viability was evaluated using 3-(4,5-dimethylthiazol-2-yl)-2,5-diphenyl tetrazolium bromide (MTT) assay 24 h after treatment with SP. Cells showed an increased proliferation following pre-treatment SP of 100 μM and 1, 10 mM (**A**). J774 cell vitality was assessed following 24 h treatment with LPS 10 µg/mL and different concentrations (0.1, 1, 10, 100 μM). SP at 1 μM and 10 μM significantly locked damage caused by LPS 10 µg/mL more than SP 0.1 μM. (**B**). Data are representative of at least three independent experiments. aaa *p* < 0.001 versus Ctr; aa *p* < 0.01 versus Ctr; b *p* < 0.05 versus LPS 10 µg/mL; bbb *p* < 0.001 versus LPS 10 µg/mL.

**Figure 2 ijms-21-03026-f002:**
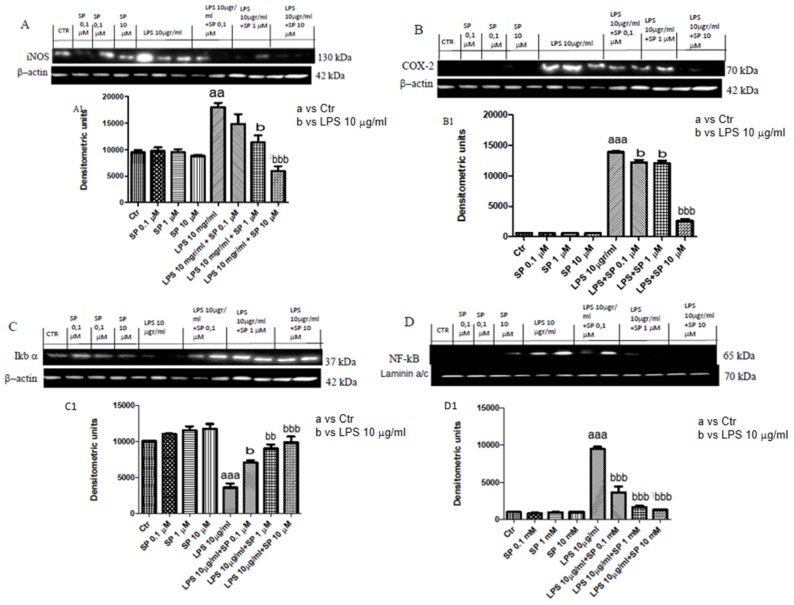
Effect of SP on the expression of iNOS, COX-2, nuclear factor of kappa light polypeptide gene enhancer in B-cells inhibitor (IκBα), and NF-κB. iNOS and COX-2 levels were increased in LPS 10 µg/mL group, whereas pre-treatment with SP at the concentration of 10 μM significantly reduced these expressions more than SP 0.1 and 1 μM (**A**,**A1**,**B**,**B1**). Blots revealed a significant increase of NF-κb expression in LPS group meanwhile its expression was attenuated in group pre-treated with SP at concentration dependent-manner (**D**,**D1**). Therefore, IκBα level was decreased in LPS, SP restored these levels at all concentrations (**C**,**C1**). Data are representative of at least three independent experiments. aa *p* < 0.01 versus Ctr; aaa *p* < 0.001 versus Ctr; b *p* < 0.05 versus LPS 10 µg/mL; bb *p* < 0.01 versus LPS 10 µg/mL; bbb *p* < 0.001 versus LPS 10 µg/mL.

**Figure 3 ijms-21-03026-f003:**
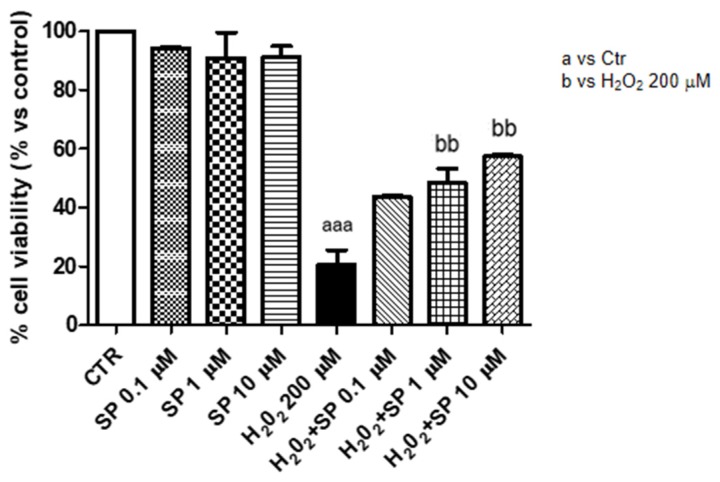
Anti-oxidant effect of SP in J774-A1 cells stimulated with H_2_O_2_. Cell viability was evaluated by MTT assay 24 h after stimulation with 200 μM H_2_O_2_. Cells showed an increased proliferation proliferative following treatment with 100 μM, 1 mM, and 10 mM SP. SP at 1 μM and 10 μM locked toxicity induced by 200 μM H_2_O_2_. Data are representative of at least three independent experiments. aaa *p* < 0.001 versus Ctr; bb *p* < 0.01 versus H_2_O_2_ 200 μM

**Figure 4 ijms-21-03026-f004:**
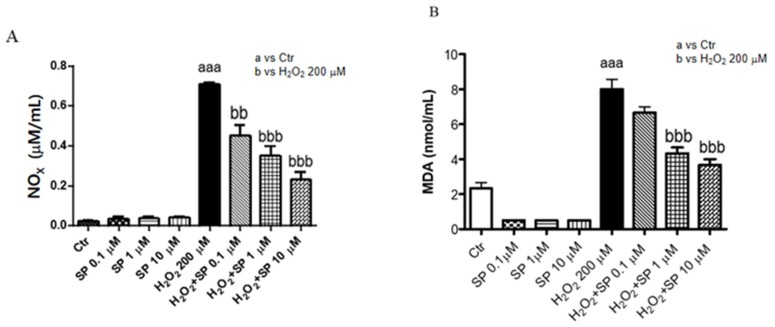
Effect of SP on nitrite production and, malondialdehyde (MDA) level. An increased of nitrite production and MDA level were evident in the 200 μM H_2_O_2_ groups, while the pre-treatment with SP at the concentration of 0.1, 1, 10 μM significantly decreased the oxidative stress-induced NOx (**A**) and MDA production (**B**). Data are representative of at least three independent experiments aaa *p* < 0.001 versus Ctr; bb *p* < 0.01 versus 200 μM H_2_O_2_; bbb *p* < 0.001 versus H_2_O_2_ 200 μM.

**Figure 5 ijms-21-03026-f005:**
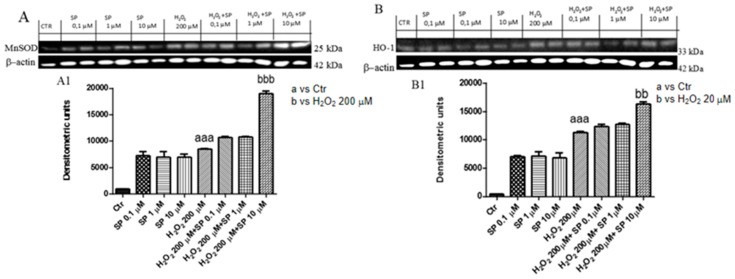
Effect of SP on antioxidant enzymes. Western blot analysis of cells lysates revealed a low increase in Mn-SOD (**A**,**A1**) and HO-1 (**B**,**B1**) levels in 200 μM H_2_O_2_ groups, whereas pre-treatment with SP 10 μM significantly restored antioxidant enzyme levels. Data are representative of at least three independent experiments. aaa *p* < 0.001 versus Ctr; bb *p* < 0.01 versus 200 μM H_2_O_2_; bbb *p* < 0.001 versus 200 μM H_2_O_2_.

**Figure 6 ijms-21-03026-f006:**
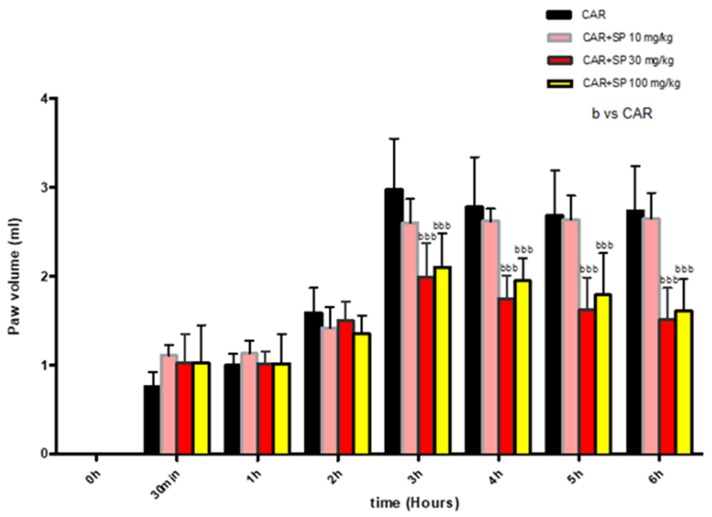
Effect of SP on time-course carrageenan (CAR)-induced paw edema. Paw edema volume was assessed at the time points indicated (*t* = 0, 30 min, 1 h, 2 h, 3 h, 4 h, 5 h, 6 h) and at different doses of SP (10, 30, and 100 mg/kg). SP groups showed significant reduction of paw volume compared to the CAR group. Values are showed as mean ± SD of 10 animals for each group. bbb *p* < 0.001 vs. CAR.

**Figure 7 ijms-21-03026-f007:**
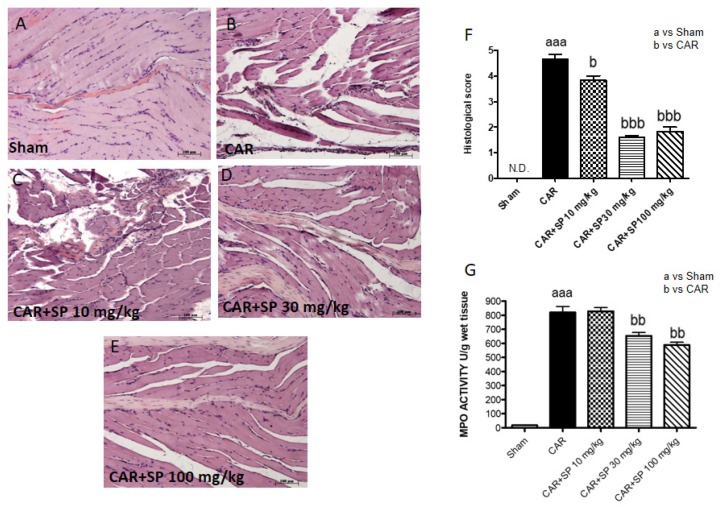
Histological analyses of paw tissues and myeloperoxidase (MPO) activity in CAR-treated rats. Control (**A**), intraplantar injection of CAR into the rat hind paw (**B**), intraplantar injection of CAR with SP 10 mg/kg (**C**), intraplantar injection of CAR with SP 30 mg/kg (**D**), and intraplantar injection of CAR with SP 100 mg/kg (**E**). Histological scores (**F**) MPO activity in paw tissues from the various treatment groups (**G**). The histological score was made by an independent observer according to this: 0 = no inflammation, 1 = mild inflammation, 2 = mild/moderate inflammation, 3 = moderate inflammation, 4 = moderate/severe inflammation, and 5 = severe inflammation. The figure is representative of at least three experiments performed on different experimental days. Values are expressed as mean SD of 10 animals for each group. aaa *p* < 0.001 vs sham; bbb *p* < 0.001 vs. CAR; bb *p* < 0.01 vs. CAR; b *p* < 0.005 vs. CAR.

**Figure 8 ijms-21-03026-f008:**
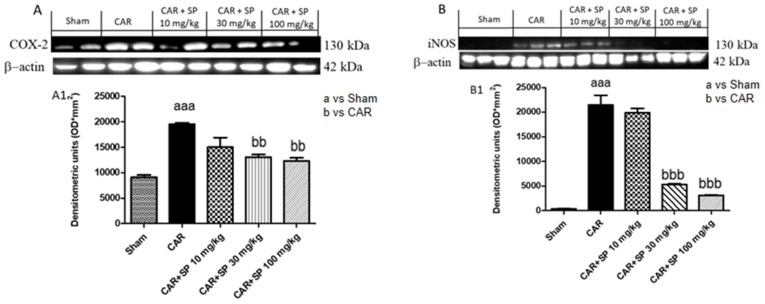
Effect of SP on expression of iNOS and COX-2 in paw tissue of CAR-treated rats evaluated by Western blot analysis. COX-2 levels in paw (**A**,**A1**) tissues were significantly increased after CAR induction; SP at 30 mg/kg and 100 mg/kg reduced COX-2 (**A**,**A1**). iNOS expression was low in paw tissue homogenates from control rats (**B**,**B1**), increased after CAR injection, and treatment with SP 30 and 100 mg/kg decrease significantly iNOS expression more than SP 10 mg/kg treatment. Data are representative of at least three independent experiments. Values are means ± SD of 10 animals for each group. aaa *p* < 0.001 vs. sham; bb *p* < 0.01 vs. CAR; bbb *p* < 0.001 vs. CAR.

**Figure 9 ijms-21-03026-f009:**
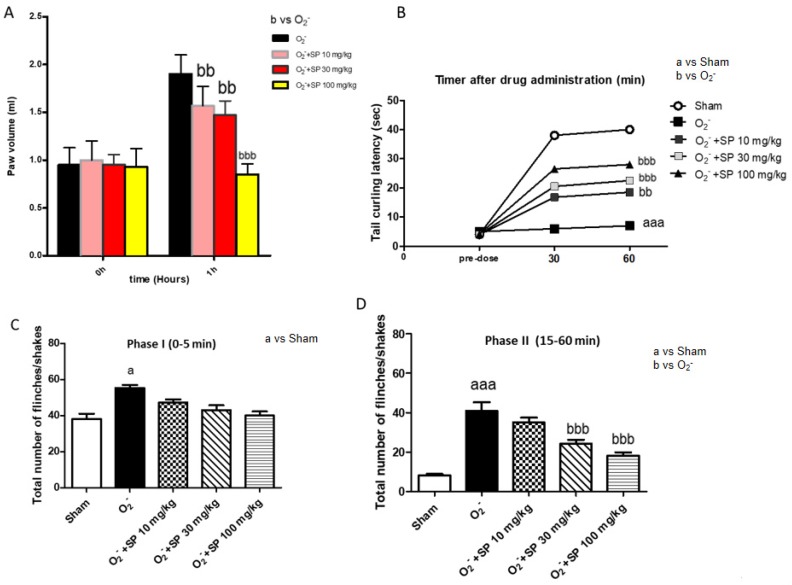
Analgesic profile of SP administration. SP when given orally 30 min before O_2_^−^ injection inhibited the paw edema volume in a dose-dependent manner, as measured at 1 h after O_2_^−^ injection (**A**). Effect of SP (10, 30, and 100 mg/kg) on nociceptive stimulus following the intraplantar injection of O_2_^−^ (**B**). The effects of systemic doses of SP (30 and 100 mg/kg) on formalin-induced pain in rats (**C**,**D**). a *p* < 0.005 versus Sham; aaa *p* < 0.001 versus. sham; bb *p* < 0.001 versus O_2_^−^ group; bbb *p* < 0.001 versus. O_2_^−^.

**Figure 10 ijms-21-03026-f010:**
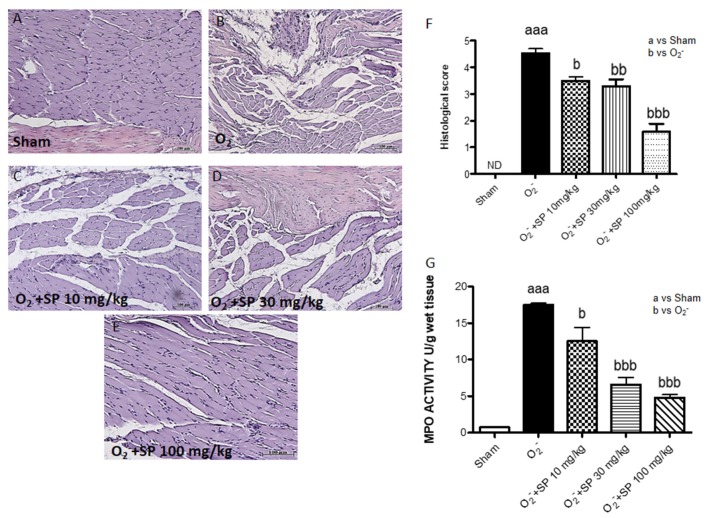
Histological evaluation of paw tissues and MPO activity following O_2_^−^ intraplantar injection. No histological alteration was observed in control group (**A**). Treatment with SP (10–30 mg/kg and 100 mg/kg) significantly reduces the pathological changes and prevents the inflammatory cells infiltration (**C**–**E**) compared to O_2_^−^ group (**B**, see histological score **F**). Effect of SP on the activity of MPO in the rat paws treated with O_2_^−^. Orally administration of SP (all doses) decreased the MPO levels after O_2_^−^-induced oxidative damage (**G**). Data is expressed as mean ± S.E.M. aaa *p* < 0.001 vs sham, b *p* < 0.05 vs. O_2_^−^; bb *p* < 0.01 vs. O_2_^−^; bbb *p* < 0.001 vs. O_2_^−^.

**Figure 11 ijms-21-03026-f011:**
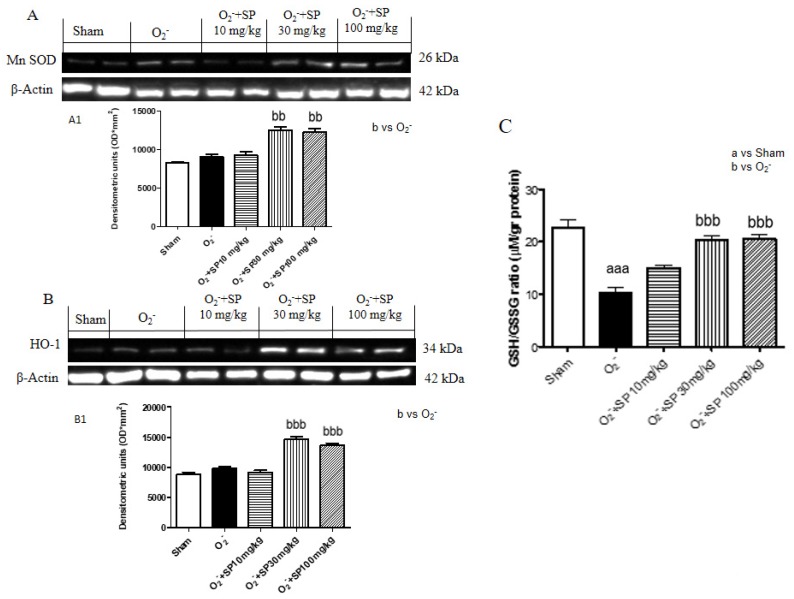
SP reduce O_2_^−^ induced oxidative stress in rat paws following O_2_^−^ intraplantar injection. Western blot analysis of hind paw tissues revealed a basal expression of MnSOD (**A**,**A1**) and HO-1 (**B**,**B1**) levels in the O_2_^−^ group, whereas oral treatment with 30 mg/kg SP significantly increased levels more than SP 100 mg/kg. GSH/GSGG ratio was measured on hind paw tissues. SP treatment (30 mg/kg and 100 mg/kg) significantly reduce the GSH/GSGG ratio (**C**) aaa *p* < 0.001 vs. sham; bb *p* < 0.01 vs. O_2_^−^; bbb *p* < 0.001 vs. O_2_^−^.

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
