# Peer review of "The Anti-Inflammatory and Antioxidant Effects of Sodium Propionate"

_ijms, 2020, doi:10.3390/ijms21083026_

Round 1

Reviewer 1 Report

Following the analysis of the manuscript titled "The anti-inflammatory and antioxidant effects of sodium propionate", I recommend that it should be revised taking into account the following observations:

  1. Please detail the full name of all abbreviated terms when first time mentioned in the abstract (i.e. LPS, CAR, NF-κB) and in the text (i.e. NF-κB).
  2. Please mention the originality element of the study, as well as its limitations.
  3. Please provide the number and data for the study’s approval.
  4. In 'author contribution' section, please specify the ones who performed statistical analysis and histological examination.
  5. Please expand the introduction and the discussion, and also enhance the references list as one single reference is not relevant for statements like:
  • ‘Several studies focused their attention on the effect of SCFA on inflammatory signaling pathways, especially, it was well demonstrated that butyrate inhibits NF- B translocation, cytokines production and prevents oxidative damage in a murine model of nephropathy [4].’
  • ‘Thus, both processes are simultaneously found in many pathological conditions [8].’
  • ‘In recent years, many studies evidenced that oxidative stress plays a crucial role in the development and propagation of inflammation [14]’
  1. A solid conclusion should be based on comparing the results with several studies with the same subject, if the latter are few or inconclusive this should be mentioned.

Author Response

Response to Reviewer 1: Comments

Following the analysis of the manuscript titled "The anti-inflammatory and antioxidant effects of sodium propionate", I recommend that it should be revised taking into account the following observations.

Point 1:  Please detail the full name of all abbreviated terms when first time mentioned in the abstract (i.e. LPS, CAR, NF-κB) and in the text (i.e. NF-κB).

Response 1. As suggested by the reviewer, we detailed the full name of abbreviated terms in both Abstract and in the main text.

Point 2: Please mention the originality element of the study, as well as its limitations.

Response 2. As suggested by the reviewer, we mentioned the originality of the study in the Introduction section and the limitations in the Conclusion section

Point 3: Please provide the number and data for the study’s approval.

Response 3. As suggested by the reviewer, the authors provided the number and data for the study approval in the “Animals” section (Section 4.1.2)

Point 4: In the 'author contribution' section, please specify the ones who performed the statistical analysis and histological examination.

Response 4. As suggested by the reviewer, the authors specified who performed the statistical analysis and histological examination

Point 5: Please expand the introduction and the discussion, and enhance the references list as one single reference is not relevant for statements like:

-‘Several studies focused their attention on the effect of SCFA on inflammatory signaling pathways, especially, it was well demonstrated that butyrate inhibits NF- B translocation, cytokines production and prevents oxidative damage in a murine model of nephropathy [4].’

-‘Thus, both processes are simultaneously found in many pathological conditions [8].’

-‘In recent years, many studies evidenced that oxidative stress plays a crucial role in the development and propagation of inflammation [14]’

Response 5. As suggested by the reviewer, we expanded the introduction and the Discussion sections as well as enhanced the reference list. 

Point 6: A solid conclusion should be based on comparing the results with several studies with the same subject, if the latter are few or inconclusive this should be mentioned.

Response 6. As suggested by the reviewer, the authors added a solid conclusion based on the previous study

Reviewer 2 Report

The paper “The anti-inflammatory and antioxidant effects of sodium propionate”  is aimed to demonstrate the effects of sodium propionate as an anti-inflammatory and antioxidants agent. The Authors performed a very wide number of different experiment both in vitro and in vivo models. In my opinion, the results they reached are very interesting, but unfortunately, the paper shows some critical points that should be addressed. The most important concern regards the approval for the in vivo study as the Authors state in section 4.2.1 that “University of Messina Review Board for the care of animals, approved this study”, but they do not give any number or other reference about the permission. On the other hand, Italian Low on animal experimentation establishes the need for an authorization from the  Italian Ministry of Health. Without this reference, the paper, in my opinion, should not be suitable for publication. Then the other issues.  Firstly, the English text should be extensively revised, as in some cases it is very hard to understand what the Authors mean. Secondly, the sections Results and Material and Methods should be better organized as in this form are sometimes difficult to follow.  Thirdly, the legend of the figured should be rephrased and some figures do not match with the description. Finally, there are lots of Greek characters mistyped (@ instead of µ or β as an example), and the references should be carefully checked as most of them are not reported in the format required by the Journal. For all these reasons the paper can be accepted only after major revisions.

Introduction

Line 35: change “exerting” as “exert”

Line 39: NF-@B ? (please check all the Greek characters throughout the paper) and also subscripts.

Line 43: spell SP

Results

General remarks:

  • I suggest dividing this section as 2.1 In vitro studies and 2.2 In vivo studies.
  • Statistical significance in the graphs plotted in the Figures could be better understood if remarked with letters instead of a symbol such as * or #.
  • The legend of the Figures should be more clear (to many information, some of them could be addressed in the text)

Line 55; change “Cellular viability..was assessed” as “In order to choose the highest SP concentrations with the lowest toxicity, cell viability was assessed stimulating cells with different concentrations (0.1 -10 -100 µM, and 1, 10 mM) of SP.”

Line 56: change “0,1” as “0.1” (please check throughout the paper)

Line 56: change “the” as “The”

Line 57: what do you mean with “resulted proliferatively”? Please rephrase the sentence. Add statistical significance in the graph (Fig 1A).

Lines 59-60: “our results…stimulation” please rephrase.

Line 97: “to the anti-oxidant …minute” please rephrase

Line 115: What does it mean (µM) after NO2 levels? In the figure 4A NO2 levels are expressed as (µM/ml). Which one is correct?

Figure 4: I suggest to divide this figure as follow: Fig 4, graphs A and B; Fig 5  graphs C and D.

Lines 132-133: add statistic in order to confirm your affirmation (also in Fig 5); When did the paw oedema start?

Line 138: “dose-dependent manner”; there is not a dose-dependence.

Line 141: spell H&E

Figure 6: the legend addresses wrong images and/or graphs

Figure 7: line 173-179: please check the whole legend.

Figure 8 line 186: “as measured at 1 h” from when or what?

Line 200: “The early analgesic response occurs..” Statistic is missing

Figure 10: the affirmation at line 239 “increased” does not match with the bar in the graph where I appreciate a decrease. Graph In 10C unit of measure of Y-axis is GSH/GSSG. The Authors should explain why (maybe in Material and Methods section)

Discussion

Lines 243-244: please rephrase the sentence “Oxidative….another”

Line 247: please rephrase the sentence “while…..documented”

Lines 250-251: please rephrase the sentence “Moreover, …..showing”

Lines 253-254: What do the Authors mean with “peripheral zones”?

Line 262; add a “a” before “relatively”

Line 263: add “it” before “is” and remove “the” before “more”

Line 262: Change “because” as “as”

Line 268: what do the Authors mean with “keep in mind”?

Line 283: remove “Z. O.”

Material and Methods

In my opinion, this section should be completely rewritten, as in this form is very difficult to follow.

Remove 4.1.2 Vital staining

4.1.4 NOX assay: not clear (add a brief explanation of the method and cell treatment)

4.1.5 As above

4.1.6 specify how many replicate in each experiment and how many experimental replicates

4.2.4 change “subjected…saline” as “rats were orally administered with…”

4.3.1 This section can be moved in 4.2.2 (paw oedema is not a behavioural test)

4.3.2 This section should be rewritten because in this form it is not clear. Specify in which groups of rats.

4.3.3 Specify in which groups of rats. Lines 417-420: this sentence is not clear, please rephrase.

4.3.4 Specify in which groups of rats.

4.3.5 Specify in which groups of rats.

From 4.3.5 until 4.3.9: these are not Behavioral examinations

4.3.8: how many replicates?

4.3.9: this section is not clear. Please rephrase.

Author Response

Response to Reviewer 2: Comments

The paper “The anti-inflammatory and antioxidant effects of sodium propionate” is aimed to demonstrate the effects of sodium propionate as an anti-inflammatory and antioxidants agent. The Authors performed a very wide number of different experiment both in vitro and in vivo models. In my opinion, the results they reached are very interesting, but unfortunately, the paper shows some critical points that should be addressed.

Point 1

A) The most important concern regards the approval for the in vivo study as the Authors state in section 4.2.1 that “University of Messina Review Board for the care of animals, approved this study”, but they do not give any number or other reference about the permission. On the other hand, Italian Low on animal experimentation establishes the need for an authorization from the Italian Ministry of Health. Without this reference, the paper, in my opinion, should not be suitable for publication.

Response A. As suggested by the reviewer, the authors provided the number and data of the authorization for animal experimentation.

B) Then the other issues. Firstly, the English text should be extensively revised, as in some cases it is very hard to understand what the Authors mean.

Response B. As suggested by the reviewer, the authors revised the English in all the text

C) Secondly, the sections Results and Material and Methods should be better organized as in this form are sometimes difficult to follow.

Response C. As suggested by the reviewer, the authors reorganized the Results and Materials and Methods sections

D) Thirdly, the legend of the figured should be rephrased and some figures do not match with the description.

Response C. As suggested by the reviewer, the figures legend were rephrased given attention to the text description.

E) Finally, there are lots of Greek characters mistyped (@ instead of µ or β as an example), and the references should be carefully checked as most of them are not reported in the format required by the Journal. For all these reasons the paper can be accepted only after major revisions.

Response E. As suggested by the reviewer, all Greek terms were adjusted and the references were reported in the correct format.

Point 2: Introduction.

A) Line 35: change “exerting” as “exert”

 Response A. As suggested by the reviewer the authors changed “exerting” with“exert”

B) Line 39: NF-@B? (Please check all the Greek characters throughout the paper) and also subscripts

Response B. As suggested by the reviewer the authors Greek characters were checked in the whole paper

C) Line 43: spell SP

Response C. As suggested by the reviewer the authors spell SP

Point 3: Results

A) General remarks: I suggest dividing this section as 2.1 In vitro studies and 2.2 In vivo studies. Statistical significance in the graphs plotted in the Figures could be better understood if remarked with letters instead of a symbol such as * or #. The legend of the Figures should be more clear (to many information, some of them could be addressed in the text).

Response A. As suggested by the reviewer, the authors divided the results in sections 2.1 “In vitro studies” and  2.2 “In vivo studies”. The statistical significance in all graphs and in the Figures legend was remarked with letters “a” and “b”. The legend of the Figures was rephrased.

B) Line 55, change “Cellular viability..was assessed” as “In order to choose the highest SP concentrations with the lowest toxicity, cell viability was assessed stimulating cells with different concentrations (0.1 -10 -100 µM, and 1, 10 mM) of SP.”

Response B. As suggested by the reviewer, we changed the phrase “Cellular viability..was assessed..” (Line 61 to 64)

C) Line 56: change “0,1” as “0.1” (please check throughout the paper)

Response C. As suggested by the reviewer, we changed 0,1 with  0.1 in the main paper

D) Line 57: what do you mean with “resulted proliferatively”? Please rephrase the sentence. Add statistical significance in the graph (Fig 1A).

Response D. As suggested by the reviewer, the sentence was rephrased as “Treatment of SP at different concentration such as 100 μM, 1 and 10 mM, markedly increased the basal proliferation of cells” (Line 62-64). The statistical significance was added in the graph of Figure 1A.

E) Lines 59-60: “our results…stimulation” please rephrase.

Response E. As suggested by the reviewer, the sentence was rephrased as “The J774-A1 cells pre-treated with SP showed an increased proliferation following LPS-induced cytotoxicity” (Line 65-66)

F) Line 97: “to the anti-oxidant …minute” please rephrase

Response F. As suggested by the reviewer, the sentence was rephrased  as “To evaluate the antioxidant effect of SP and its potential capability to induce recovery after oxidative stress, J774-A1 cells were pre-treated with SP and then stimulated with H2O2 200 mM for 10 minutes” (Line 101 to 103)

G) Line 115: What does it mean (µM) after NO2 levels? In the figure 4A NO2 levels are expressed as (µM/ml). Which one is correct?

Response G. The correct unit of measurement is µM/ml

H) Figure 4: I suggest to divide this figure as follow: Fig 4, graphs A and B; Fig 5 graphs C and D.

Response H. As suggested by the reviewer, the authors divided Figure 4 and Figure 5.

I) Lines 132-133: add statistic in order to confirm your affirmation (also in Fig 5); When did the paw oedema start?

Response I. As suggested by the reviewer, we added the statistic in the sentence at Line 144 and in the legend of Figure 6. The paw edema started at 3 h until 6 h (Line 144)

 J) Figure 8 line 186: “as measured at 1 h” from when or what?

Response J. As suggested by the reviewer, the legend of Figure 9 was rephrased “SP when given orally 30 min before O2- injection inhibited the paw edema volume in a dose-dependent manner, as measured at 1 h after O2- injection” (Line 196 to 200).

K) Line 200: “The early analgesic response occurs” Statistic is missing.

Response K. As suggested by the reviewer, the statistic was added as “*P < 0.005 versus Sham” (Line 209)

L) Figure 10: the affirmation at line 239 “increased” does not match with the bar in the graph where I appreciate a decrease. Graph In 10C unit of measure of Y-axis is GSH/GSSG. The Authors should explain why (maybe in the Material and Methods section)

Response L. As suggested by the reviewer, we corrected the sentence under Figure 11 (Line 237-238). Moreover, the authors explained the measurement of GSH/GSGG in the 4.2.11 section (Line 47 to 475)

Point 4: Discussion

A) Lines 243-244: please rephrase the sentence “Oxidative….another”

Response A. As suggested by the reviewer, the sentence was rephrased as “Inflammation and oxidative stress are linked together in a large number of pathophysiological processes” (Line 249-250).

B) Line 247: please rephrase the sentence “while…..documented”

Response B. As suggested by the reviewer, the sentence was changed as “while the molecular mechanism of propionate is not well documented” (Line 252).

C) Lines 250-251: please rephrase the sentence “Moreover, …..showing”

Response C. As suggested by the reviewer, the phrase was changed as “moreover, we have previously shown the neuroprotective effect of SP in in vitro in neuroinflammatory model and in vivo model of spinal cord injury (SCI, recognizing in SP an optimal therapeutic target for neuroinflammatory disorders. In this study, we confirmed the ability of SP to contain the peripheral acute inflammation trough the down-regulation of NF-κB pathway” (Line 255 to 258)

D) Lines 253-254: What do the Authors mean with “peripheral zones”?

Response D. The authors meant “peripheral acute inflammation” (Line 257)

Point 5. Material and Methods

A) In my opinion, this section should be completely rewritten, as in, this form is very difficult to follow.

Response A. As suggested by the reviewer, the Material and Methods section was revised.

B) 4.1.4 NOX assay: not clear (add a brief explanation of the method and cell treatment)

Response B. As suggested by the reviewer, the authors explained the section 4.1.3

C) 4.1.5 as above

Response C. As suggested by the reviewer, the authors explained the section 4.1.4

D) 4.1.6 specify how many replicate in each experiment and how many experimental replicates

Response C. As suggested by the reviewer, the authors specified the numbers of replicates in the section 4.1.5 “Each analysis was performed three times with three samples replicates for each one”

E) 4.2.4 change “subjected…saline” as “rats were orally administered with…”

Response E. As suggested by the reviewer, the authors changed the phrases

F) 4.3.2 this section should be rewritten because in this form it is not clear. Specify in which groups of rats.

Response D. As suggested by the reviewer, the Tail-flick test section (4.2.6.1.) was rewritten

G) 4.3.3 Specify in which groups of rats. Lines 417-420: this sentence is not clear, please rephrase. 4.3.4 Specify in which groups of rats. 4.3.5 Specify in which groups of rats.

Response E. As suggested by the reviewer, the authors specified the group of rats in the sections: 4.2.6.1- 4.2.6.2. and 4.2.7.

H) 4.3.8: how many replicates?

Response F. Data are representative of at least three replicates

I) 4.3.9: this section is not clear. Please rephrase.

Response G. As suggested by the reviewer, section 4.2.11. was rewritten

Round 2

Reviewer 2 Report

This version of the  “The anti-inflammatory and antioxidant effects of sodium propionate”  has been extensively revised and improved by the Authors. However, some minor point should be still addressed.   The most important concern remains the approval for the in vivo study: the Authors provides in the new version the protocol number of permission by the University of Messina Review Board for the care of animals but is still missed the authorization from the  Italian Ministry of Health that is compulsory according to the Italian Legislation (DL 26/2014).

Abstract

Line 18; 22 and 24: maybe the term “following” can be changed in “after”

Introduction

Line 50: remove “previously”

Results

General remark: the different letters on the bars of the graphs explain statistical differences among groups. An example: Figure 1B –  a = control; if LPS treated group is statically different from control should be indicated as b; if SP 0,1 is different from LPS but not statistically different from control, should be indicated as a as well…and so on. The same for all the graphs.

Line 66: change “following” with “after”

Line 109: change “treatment” with “pre-treatment

Line 144: remove “that increase”

Discussion

Line 255: change “shown” as “demonstrated”

Line 273: change “highlighting” in “highlight”

Material and Methods

Line 350: remove “E” after “Talero”

References

References: 1; 2; 3; 4; 6; 8; 9; 10; 12; 15; 17; 19; 21-25; 27-30; 32-34 are not addressed following the Instruction for Authors of the Journal.

Please note the number of references must not exceed 200 for any article.

Journal Paper

Use the sequence: [Author surname] [Author initials], [Other author surnames & initials]. [Article title]. [Journal name abbreviation]. [Year]; [Volume]: [First page number]-[Last page number].

Journal paper must have journal name (or journal abbreviation), year, volume and page numbers.

Please make sure that the year of publication is not missing.

Please use ":" to separate volume and first page number.

Do not bold or italicize the title, journal name, or any part of references. Omit any "." in the journal abbreviations.

Examples:

  1. Eknoyan G, Beck GJ, Cheung AK, et al. Effect of dialysis dose and membrane flux in maintenance hemodialysis. N Engl J Med. 2002; 347: 2010-9.

Supplement example:

  1. Volk HD, Reinke P, Krausch D, et al. Monocyte deactivation-rationale for a new therapeutic strategy in sepsis. Intensive Care Med. 1996; 22 (Suppl 4):S474-S481.

No author given example:

  1. [No authors listed]. Medicare program; criteria for Medicare coverage of adult liver transplants-HCFA. Final notice. Fed Regist. 1991; 56(71):15006-15018.

In press example:

  1. Cheung TMT, et al. Effectiveness of non-invasive positive pressure ventilation in the treatment of acute respiratory failure in severe acute respiratory syndrome. Chest; in press.

Epub ahead of print example:

  1. Li W, Chen Y, Cameron DJ, et al. Elovl4 haploinsufficiency does not induce early onset retinal degeneration in mice. Vision Res 2007; [Epub ahead of print].

Book

Kiloh LG, Smith JS, Johnson GF, et al. Physical treatment in psychiatry. Boston, USA: Blackwell Scientific Publisher; 1988.

Chapters in Edited Book

Beckenbough RD, Linscheid RL. Arthroplasty in the hand and wrist. In: Green DP, ed. Operative Hand Surgery, 2nd ed. New York: Churchill Livingstone; 1988: 167-214.

Web Site (discouraged)

[Internet] WHO. Summary of probable SARS cases with onset of illness from 1 November 2002 to 31 July 2003. http://www.who.int/

[Internet] Kornberg R. http://nobelprize.org/

DOI: Please remove the "[pii]" field in any doi. E.g., doi:00932.2006 [pii] 10.1152/japplphysiol.00932.2006. Remove "00932.2006 [pii]".